# Scheduling jobs with stochastic holding costs

**Dabeen Lee**
Discrete Mathematics Group
Institute for Basic Science (IBS)
Daejeon, South Korea
dabeenl@ibs.re.kr

**Milan Vojnovic**
Department of Statistics
London School of Economics
London, United Kingdom
m.vojnovic@lse.ac.uk

## Abstract

This paper proposes a learning and scheduling algorithm to minimize the expected cumulative holding cost incurred by jobs, where statistical parameters defining their individual holding costs are unknown a priori. In each time slot, the server can process a job while receiving the realized random holding costs of the jobs remaining in the system. Our algorithm is a learning-based variant of the $c\mu$ rule for scheduling: it starts with a preemption period of fixed length which serves as a learning phase, and after accumulating enough data about individual jobs, it switches to nonpreemptive scheduling mode. The algorithm is designed to handle instances with large or small gaps in jobs' parameters and achieves near-optimal performance guarantees. The performance of our algorithm is captured by its regret, where the benchmark is the minimum possible cost attained when the statistical parameters of jobs are fully known. We prove upper bounds on the regret of our algorithm, and we derive a regret lower bound that is almost matching the proposed upper bounds. Our numerical results demonstrate the effectiveness of our algorithm and show that our theoretical regret analysis is nearly tight.

## 1 Introduction

We consider the following algorithmic question: given a list of jobs, each of which requires a certain number of time steps to be completed while incurring a random cost in every time step until finished, learn the relative priorities of jobs and make scheduling decisions of which job to process in each time step, with the objective of minimizing the expected total cumulative cost. Here, we need an algorithm that seamlessly integrates learning and scheduling.

This question is motivated by several applications. Online social media platforms moderate content items, and here, jobs correspond to reviewing content items while the jobs have random costs driven by the number of accumulated views as views of harmful content items represent a community-integrity cost. Then the content review jobs are prioritized based on popularity of content items [18]. More generally, modern data processing platforms handle complex jobs whose characteristics are often unknown in advance [10], but as a system learns more about the jobs' features, it may flexibly adjust scheduling decisions to serve jobs with high priority first. Another application is in optimizing energy consumption of servers in data centers, where a job waiting to be served uses energy-consuming resources [3, 7]. In emergency medical departments, patients undergo triage while being treated, and schedules for serving patients are flexibly adjusted depending on their conditions [14, 15]. Note that patients' conditions may get worse while waiting, which corresponds to holding costs in our problem. For aircraft maintenance, diagnosing the conditions of parts and applying the required measures to repair them are conducted in a combined way [1].

We model and study the problem as a single-server scheduling system where job holding costs are according to stochastic processes with independent and identically distributed increments with unknown mean values, under the assumption that their mean service times are known. Recent works,

35th Conference on Neural Information Processing Systems (NeurIPS 2021).

e.g. [5], started investigating queuing system control policies under uncertainty about jobs' mean service time parameters, but restricting to the setting where job holding costs are assumed to be deterministic (linear) functions of stochastic job waiting times. In our problem setting, job holding costs are stochastic in a different way in that job holding costs themselves are according to some stochastic processes. Under the aforementioned application scenarios, it is more natural to model a job's holding costs by a stochastic process. As the first step towards understanding the setting of stochastic job holding costs, we consider a single-server scheduling with a fixed list of jobs.

Since the statistical parameters quantifying the random holding costs of jobs are unknown to the decision-maker, the main challenge is to learn the parameters of jobs, thereby obtaining a near-accurate priority ranking of jobs to find a scheduling policy minimizing the total accumulated cost.

**Problem formulation**

We consider a discrete-time scheduling with a single server and $N$ jobs. The jobs, labelled by $i = 1, \ldots, N$, are present in the system from the beginning, and we assume no further job arrivals. Every job remaining in a time slot incurs a random holding cost at the beginning of the time step, and holding costs incurred by a job are according to a sub-Gaussian distribution and i.i.d. across time slots. Sub-Gaussian distributions allow for different parametric distributions, e.g. Bernoulli and Poisson, which are suitable for modeling jobs incurring random costs incrementally. The mean holding costs of jobs are given by $c_1, \ldots, c_N$ whose values are a-priori unknown. The number of required time steps to complete job $i$ is given deterministically by $T/\mu_i$ for each $i$, where $T$ is a scaling parameter. Note that the larger the value of $T$ the larger the number of per-job stochastic cost observations. The larger value of $T$ does not necessarily mean that mean job service times are large in real time. In addition to the case of deterministic job service times, we also consider the case of stochastic job service times, assumed to be according to geometric distributions, which is a standard case studied in the context of queuing systems.

We analyze the performance of our scheduling policy against the minimum (expected) cumulative holding cost that can be achieved when the jobs' mean holding cost values are fully known. The famous $c\mu$ *rule*, which sequentially processes jobs in the decreasing order of their $c_i\mu_i$ values, is known to guarantee the minimum cumulative holding cost, so we use this as our benchmark. Assuming $c_1\mu_1 \geq c_2\mu_2 \geq \cdots \geq c_N\mu_N$, we can define the *regret* of a scheduling algorithm $\varphi$ as

$$R(T) := C(T) - \sum_{i \in [N]} c_i \sum_{j \in [i]} T/\mu_j$$

where $C(T)$ is the expected cumulative holding cost of $\varphi$, i.e., $C(T) = \sum_{i \in [N]} c_i T_i^\varphi$ with $T_i^\varphi$ denoting the completion time of job $i$ and the second term on the right-hand side is the minimum expected cumulative holding cost, achievable by the $c\mu$ rule. Note that $R(T)$ is a random variable, as it may depend on randomness in the algorithm $\varphi$. We will consider $\mathbb{E}[R(T)]$, the *expected regret* of the algorithm. When $c_i$'s and $\mu_i$'s are constants, the minimum cumulative holding cost is $O(N^2 T)$. Here, our goal is to construct a policy whose expected regret is sublinear in $T$ and subquadratic in $N$.

Previous work [5] has focused on the case when the values of $\mu_1, \ldots, \mu_N$ are unknown while the values of $c_1, \ldots, c_N$ are known. As the service time of a job is instantiated only after completing it, this work relies on assumptions that there is a fixed list of classes and multiple jobs may belong to the same class, which means that each $\mu_i$ can take one of a few values. In contrast, our paper is focused on addressing the challenge of uncertain costs. Our framework allows jobs to differ from each other (equivalently, each job corresponds to a unique class), but to make learning distinct jobs possible, we assume that the factor $T$ in the service times is large. We can think of the large $T$ regime as the case where the jobs' features are observed frequently, as well as the setting of long job service times.

**Our contributions**

We develop our algorithm based on the *empirical $c\mu$ rule*, that is, the $c\mu$ rule applied with the current estimates of the mean holding costs. Since the ranking of jobs based on the $\hat{c}_{i,t}\mu_i$ values, where $\hat{c}_{i,t}$ denotes the empirical mean of job $i$'s holding cost in time slot $t$, may change over time, it is natural to come up with two types of the empirical $c\mu$ rule, preemptive and nonpreemptive. Under the *preemptive* empirical $c\mu$ rule, the server selects a job for every single time slot. In contrast, under the *nonpreemptive* version, once a job is selected in a certain time slot, the server has to commit to

serving this job until its completion, and then, it may select the next job based on the empirical $c\mu$ rule. The preemptive empirical $c\mu$ rule works well for instances with large gaps between the jobs' mean holding costs, whereas the nonpreemptive one is better for cases where the jobs' mean holding costs are close. However, it turns out that, under both the preemptive and nonpreemptive cases, the expected regret can grow linearly in $T$ in the worst case. The preemptive case may result in undesired delays especially for jobs with similar values, while the nonpreemptive case may suffer from early commitment to a job with low priority.

Our policy, the *preemptive-then-nonpreemptive* empirical $c\mu$ rule, is basically a combination of the preemptive and nonpreemptive empirical $c\mu$ rules. This variant of empirical $c\mu$ rule has a fixed length of preemption phase followed by nonpreemptive scheduling of jobs. The preemption period is long enough to separate jobs with large gaps in their mean holding costs, while it is not too long so that we can control delay costs from the preemption phase to be small, thereby avoiding undesired delays from continuous preemption and the risk of early commitment.

In Section 3, we give a theoretical analysis of our algorithm for the case of deterministic service times. We prove that the expected regret of our empirical $c\mu$ rule is sublinear in $T$ and subquadratic in $N$. We also show that this is near-optimal by providing a lower bound on the expected regret of any algorithm that has the same scaling in $T$ while there is a small gap in terms of the dependence on $N$. Furthermore, we give instance-dependent upper bounds that delineate how our algorithm performs depending on the gaps between $c_i\mu_i$ values. In Section 4, we consider the setting where the service time of each job is stochastic and geometrically distributed. Our analysis reveals that the expected regret of our algorithm for the case of geometrically distributed service times is also sublinear in $T$ and subquadratic in $N$. Lastly, in Section 5, we discuss results from numerical experiments that demonstrate our algorithm's performance and support our theoretical claims.

## 2   Related work

The scheduling problem asking to minimize the sum of weighted completion times for a given set of jobs, with weights $c_i$ and processing times $1/\mu_i$, was studied in the seminal paper by Smith [13], who showed that serving jobs in decreasing order of indices $c_i\mu_i$ is optimal. This policy is often referred to as the *Smith's rule*. This policy corresponds to the weighted shortest processing time first (WSPT) policy in the literature on machine scheduling. The Smith's rule is also optimal for the objective of minimizing the expected sum of weighted completion times when job processing times are random with mean values $1/\mu_i$. We refer the reader to [11] for a comprehensive coverage of various results.

Serving jobs by using $c_i\mu_i$ as the priority index is known to be an optimal scheduling policy for *multi-class, single-server queuing systems*, with arbitrary job arrivals and random independent, geometrically distributed job processing times with mean values $1/\mu_i$ [2]. This policy is commonly referred to as the $c\mu$ *rule*. A generalized $c\mu$ rule is known to be asymptotically optimal for convex job holding cost functions in a heavy-traffic limit, where the generalization corresponds to using a dynamic index defined as the product of the current marginal job holding cost and the mean processing time of a job [17]. This generalized $c\mu$ rule is also known to be asymptotically optimal in a heavy-traffic limit for *multi-server* queuing systems under a certain resource polling condition [9].

The work discussed above on the performance of Smith's or $c\mu$ rule assumes that the values of marginal job holding costs and mean processing time parameters are known to the scheduler. Only some recent work considered the performance of these rules when some of these parameters are unknown. In the line of work on *scheduling with testing* [6, 15], marginal job holding costs and mean processing times of jobs are a-priori unknown, but their values for a job become known by *testing* this job. The question there is about how to allocate the single server to processing and testing, which cannot be done simultaneously. The optimal policy combines testing the jobs up to certain time and serving the jobs according to the $c\mu$ rule policy. In [5], a multi-class queuing system is considered under the assumption that mean job processing times are unknown to the scheduler. The authors established that using the empirical $c\mu$ rule in the single-server case guarantees a finite regret with respect to the $c\mu$ rule with known parameters as a benchmark. Similar result is established for the multi-server case by using the empirical $c\mu$ rule combined with an exploration mechanism. Unlike [5], in our setting, each job corresponds to a unique class, so an exploration phase is required for each job. The preemption phase in our algorithm is for exploration within each job.

Finally, our work is related to *permutation* or *learning to rank* problems, e.g. see [4, 8] and the references therein, where the goal is to find a linear order of items based on some observed information about individual items, or relations among them. Indeed, the objective of our problem can be seen as finding a permutation $\pi$ that minimizes the cost function $\sum_{i=1}^{N} ic_{\pi(i)}$, for the special case of identical mean processing time parameter values. For example, we may interpret $c_i$ as a measure of dissimilarity between item $i$ and a reference item, and the goal is to sort items in decreasing order of these dissimilarity indices. The precise objective is defined for a sequential learning setting where irrevocable ranking decisions for items need to be made over time and the cost in each time step is the sum of dissimilarity indices of items which are still to be ranked.

## 3 Algorithm and regret bounds

Before we formally describe our algorithm, we consider the case of $N = 2$ and $\mu_1 = \mu_2 = 1$ for motivation. Without loss of generality, let us assume that $c_1 \geq c_2$. Then processing job 1 first and job 2 next is optimal, and the minimum expected cost is $c_1 T + c_2 \cdot 2T$. Recall that the empirical $c\mu$ rule selects whichever job that has a higher estimated mean holding cost while the preemptive and nonpreemptive versions differ in how frequently such selections are made.

The preemptive version is more flexible in that scheduling decisions may be adjusted in every time slot as the empirical estimates of $c_1$ and $c_2$ are updated. This is indeed favorable when $c_1$ is much greater than $c_2$, in which case, the empirical estimate of $c_1$ would get significantly larger than that of $c_2$ soon. However, we can imagine a situation where $c_1$ and $c_2$ are so close that the empirical estimates of $c_1$ and $c_2$ are almost identical for the entire duration of processing the jobs. Under this scenario, the two jobs are chosen with almost equal probabilities, in which case, they are completed around the same time. For example, job 1 stays in the system for $2T$ time periods, while job 2 remains for $2T - 1$ time steps. Then the regret is $c_1 2T - c_2$, which may be linear in $T$.

The issue is that both jobs may remain in the system and incur holding costs for the entire duration of service $2T$. In contrast, one job leaves the system after $T$ time steps under the optimal policy. Therefore, there is an incentive in completing one job early instead of keeping both jobs longer.

Inspired by this, we could consider the nonpreemptive version that selects a job in the beginning and commits to it. However, under the nonpreemptive version, the probability of job 2 being selected first is at least $(1 - c_1)c_2$, and therefore, the expected regret of this policy is at least $(1 - c_1)c_2 \cdot (c_1 - c_2)T$ as $(c_1 - c_2)T = (c_1 \cdot 2T + c_2 T) - (c_1 T + c_2 \cdot 2T)$. When $c_1 - c_2 = \Omega(1)$, the expected regret is linear in $T$. Hence, the nonpreemptive version may suffer from undesired early commitment.

Our algorithm, which we call the preemptive-then-nonpreemptive empirical $c\mu$ rule, is a combination of the above two policies. As the preemptive version, we start with a preemption phase in which the server may try different jobs while learning the mean holding costs of jobs, thereby circumventing the early commitment issue. At the same time, we limit the number of preemption steps, which allows avoiding the issue of letting job 1 to stay in the system undesirably long for the case of 2 jobs.

---

**Algorithm 1** Preemptive-then-nonpreemptive empirical $c\mu$ rule

---

**Require:** $T_s$ is a predetermined length of the preemption period. $J \leftarrow [N]$.
    **(Preemption phase)** For each time step $t \in [T_s]$, select and serve a job in $\arg\max_{i \in [N]} \hat{c}_{i,t}\mu_i$.
    **(Nonpreemptive phase)** While $J$ is nonempty, select and complete a job $j$ in $\arg\max_{i \in J} \hat{c}_{i,t}\mu_i$
    where $t$ is the current time step index. Then $J \leftarrow J \setminus \{j\}$.

---

After the preemption period, we process jobs nonpreemptively, which means that once a job is chosen, we commit the server to the job until it is done. If a job has a significantly higher empirical estimate than the others, then we can easily choose the job. Even if not, we still stick to a job regardless of whether the job turns out to be of lower cost than others later. Our algorithm's performance heavily depends on the length of the preemption phase, denoted $T_s$, which we will decide later in this section.

### 3.1 Regret upper and lower bounds

Recall that a random variable $X$ with mean $c$ is sub-Gaussian with parameter $\sigma$ if $\mathbb{E}[X^{\lambda(X-c)}] \leq \exp(\sigma^2 \lambda^2 / 2)$ for all $\lambda \in \mathbb{R}$. Note that if $X$ is sub-Gaussian with parameter $\sigma$, then for any $\sigma' \geq \sigma$, it

is sub-Gaussian also with parameter $\sigma'$. Moreover, if $X$ is sub-Gaussian with parameter $\sigma$, then $X/\beta$ is sub-Gaussian with parameter $\sigma/\beta$ for any $\beta > 0$. As the total holding cost depends linearly on $c_1, \ldots, c_N$, we may assume that $c_i \in [0,1]$ and $\sigma_i = 1$ for $i \in [N]$ without loss of generality.

**Special warm-up case: two jobs**  We start with the setting of $N = 2$ and $\mu_1 = \mu_2 = 1$. When $c_1$ and $c_2$ are almost identical, Algorithm 1 would spend almost equal numbers of time steps for the jobs during the preemption phase. This means that the job with the higher holding cost is delayed for $T_s/2$ time steps, in which case, we can argue that the (expected) regret is linear in $T_s$. Hence, to attain a regret upper bound that is sublinear in $T$, we need to set $T_s$ to be sublinear in $T$. On the other hand, we would want to have a sufficiently long period for learning the jobs to avoid the issue of early commitment to the job of less priority. In fact, setting

$$T_s = \kappa \cdot T^{2/3}(\log T)^{1/3}$$

for some constant $\kappa > 0$ minimizes the expected regret when $N = 2$ and $\mu_1 = \mu_2 = 1$.

**Theorem 3.1.** *The expected regret of Algorithm 1 for 2 jobs is $O(T_s + T\sqrt{\log(T/T_s)})$. When $T_s = \Theta(T^{2/3}(\log T)^{1/3})$, the expected regret is $O(T^{2/3}(\log T)^{1/3})$.*

Let us give a proof sketch here. The worst case that gives rise to the regret upper bound in Theorem 3.1 occurs when $c_1$ and $c_2$ are so close that the algorithm selects job 2 after the preemption phase with a nonnegligible probability, in which case, job 2 is completed first. In such case, job 1 stays in the system for the entire duration of $2T$ time steps, while job 2 may spend up to $T_s + T$ time slots depending on how many times job 1 is chosen during the preemption phase, which results in delays for job 2. Then the corresponding regret is $(c_1 \cdot 2T + c_2(T_s + T)) - (c_1 T + c_2 \cdot 2T) = c_2 T_s + (c_1 - c_2)T$. Here, $c_1 - c_2$ is sufficiently small, for otherwise, $\hat{c}_{1,T_s+1}$ would be significantly larger than $\hat{c}_{2,T_s+1}$ with high probability.

It turns out that there is a matching lower bound on the expected regret of any policy (up to a $\log T$ factor), implying in turn that Algorithm 1 with $T_s = \Theta(T^{2/3}(\log T)^{1/3})$ is asymptotically optimal.

**Theorem 3.2.** *There is a family of instances, under which the expected regret of any fixed (randomized) scheduling algorithm is $\Omega(T^{2/3})$, where the expectation is taken over the random choice of an instance and the randomness in holding costs and the algorithm.*

Our proof of Theorem 3.2 is similar in spirit to the standard argument for proving regret lower bounds for the stochastic multi-armed bandit (MAB) problem. Basically, when the mean holding costs of two jobs are very close, we can argue based on the Kullback–Leibler (KL) divergence that no algorithm can tell which job has the higher mean holding cost with high probability for the first few time steps (see [12]). The difference is that every job incurs a cost simultaneously, unlike the MAB setting where only the arm played generates a reward, and that jobs leave after some number of time slots.

**General case: arbitrary number of jobs**  We are interested in the setting where $T$ is large and the asymptotic regret analysis of Algorithm 1 with respect to $T$. Hence, we focus on the case where $\mu_1, \ldots, \mu_N$ are fixed constants, but the following condition is sufficient for our analysis.

**Assumption.** $\mu_i \leq (T/\log(NT))^{1/3}/2$ for $i \in [N]$.

Up to scaling, we may also assume that $\mu_1, \ldots, \mu_N \geq 1$ without loss of generality. We further assume that $T/\mu_i$ for $i \in [N]$ are all integers. If $T/\mu_i$ is not an integer for some $i$, one may assume that the service time of job $i$ is $\lceil T/\mu_i \rceil$ since job $i$ needs "at least" this many time steps to be completed.

For the general case, we set the length $T_s$ of the preemption period to

$$T_s = \kappa \cdot (T/\mu_{\min})^{2/3}(\log(NT/\mu_{\min}))^{1/3}$$

for some constant $\kappa > 0$ where $\mu_{\min} := \min\{\mu_i : i \in [N]\}$. We assume that $N$ is small enough so that $T_s = o(T)$. Moreover, by the assumption, we may assume that $T_s \leq T/2\mu_i$ for all $i$.

Let us briefly elaborate on why nonpreemptive scheduling makes sense after the preemption period even for the general case. We explained in the previous section that for the case of two jobs, there is an incentive to finish one job early because keeping both jobs long results in a large regret. The same idea applies when there are many jobs, so after the preemption period, we choose a job and complete it. After finishing the first job, denoted $k$, each of the remaining $N-1$ jobs would have

spent at least $T/\mu_k - T_s \geq T/2\mu_k$ time steps in the system, from which we would have collected "enough data" to learn the mean holding costs of the jobs. Here, what we mean by enough data is that (we will prove later that) the absolute error in the estimation of each job's mean holding cost is $O(1/\sqrt{T})$ with high probability, so the regret while serving the other $N-1$ jobs scales as the rate of $O(\sqrt{T})$, which is less than $T^{2/3}$. Hence, the dependence on $T$ becomes not as critical, in contrast to the period for serving the first job. Then it is intuitive to complete one job by one as soon as possible so that we can quickly reduce the number of remaining jobs in the system. We will explain these details while analyzing the expected regret of Algorithm 1. Let $\mu_{\max} = \max\{\mu_i : i \in [N]\}$.

**Theorem 3.3.** *The expected regret of the preemptive-then-nonpreemptive empirical $c\mu$ rule is*

$$O\left(N(T/\mu_{\min})^{2/3}\left(\log(NT/\mu_{\min})\right)^{1/3} + \sqrt{\mu_{\max}/\mu_{\min}}N^{3/2}(T/\mu_{\min})^{1/2}\left(\log(NT/\mu_{\min})\right)^{1/2}\right).$$

In particular, when $\mu_1, \ldots, \mu_N$ are fixed constants, the expected regret is

$$\mathbb{E}\left[R(T)\right] = \tilde{O}\left(\max\left\{NT^{2/3}, \ N^{3/2}T^{1/2}\right\}\right) \tag{1}$$

where $\tilde{O}$ hides additional logarithmic factors in $NT$.

Our argument is more involved for the general case than the case of two jobs. We argue that the preemption phase results in $O(NT_s)$ regret due to job delays. Moreover, we analyze the regret due to the possibility that the algorithm finishes the jobs in a completely different order from the optimal sequence. Nevertheless, we use a technical lemma to compare the optimal cost and the cumulative cost from any sequence, and the key idea is that whenever a pair of two jobs is reversed, we can argue that the mean holding costs of the two jobs are close.

It turns out that this regret upper bound is near-optimal. We next provide a lower bound on the regret of any algorithm that matches the upper bound up to a small gap in the dependence on $N$.

**Theorem 3.4.** *For any (randomized) scheduling algorithm, there is a family of instances under which*

$$\mathbb{E}\left[R(T)\right] = \Omega\left(\max\left\{\mu_{\min}^{1/3}\mu_{\max}^{-5/3}N^{2/3}T^{2/3}, \ \mu_{\min}\mu_{\max}^{-3/2}NT^{1/2}\right\}\right)$$

*where the expectation is taken over the random choice of an instance and the randomness in holding costs and the algorithm.*

By Theorem 3.4, when $\mu_1, \ldots, \mu_N$ are constants, there exists a family of instances under which

$$\mathbb{E}\left[R(T)\right] = \Omega\left(\max\left\{N^{2/3}T^{2/3}, \ NT^{1/2}\right\}\right) \tag{2}$$

for any algorithm. Between (1) and (2), there are small gaps with respect to the dependence on $N$.

Our proof of Theorem 3.4 is similar to the KL divergence-based argument for the MAB problem, but the key difference is that we observe the realized costs of all jobs, unlike the MAB setting where we observe the reward of at most one arm. Moreover, jobs leave the system at some points under our setting, and we also need to deal with the costs incurred by all jobs currently in the system.

## 3.2 Instance-dependent regret upper bounds

The upper and lower bounds on the expected regret in the previous section are independent of the values of $c_1, \ldots, c_N$. However, it is intuitive to expect that Algorithm 1's performance depends on the gaps between the values of $c_1\mu_1, \ldots, c_N\mu_N$, as it would be difficult to separate jobs $i$ and $j$ with $c_i\mu_i$ and $c_j\mu_j$ being close. Motivated by this, we give regret upper bounds that have an explicit dependence on the gaps between $c_1\mu_1, \ldots, c_N\mu_N$.

For the case of $N = 2$, the expected regret depends on the quantity $\Delta$, defined as,

$$\Delta := |c_1\mu_1 - c_2\mu_2|/(\mu_1 + \mu_2)$$

that captures the gap between $c_1\mu_1$ and $c_2\mu_2$.

**Theorem 3.5.** *When $N = 2$ and $\mu_{\min} = \min\{\mu_1, \mu_2\}$, the expected regret of Algorithm 1 is*

$$\mathbb{E}\left[R(T)\right] = O\left(\frac{1}{\Delta^2}\log(T/\mu_{\min})\right).$$

Note that the expected regret scales with a rate of $\Delta^{-2}$ but the dependence on $T$ is logarithmic for fixed $\Delta$. For the general case, we consider the gap between $c_1\mu_1$ and the other $c_i\mu_i$ values. Let $\Delta_{\min}$ and $\Delta_{\max}$ be defined as follows:

$$\Delta_{\min} := \min_{i \neq 1}\left\{|c_1\mu_1 - c_i\mu_i|/(\mu_1 + \mu_i)\right\}, \quad \Delta_{\max} := \max_{i \neq 1}\left\{|c_1\mu_1 - c_i\mu_i|/(\mu_1 + \mu_i)\right\}.$$

**Theorem 3.6.** *For general $N$, the expected regret of Algorithm 1 is*

$$\mathbb{E}\left[R(T)\right] = O\left(\left(\frac{1}{\Delta_{\min}^2}N + \frac{\mu_{\max}}{\mu_{\min}}\cdot\frac{\Delta_{\max}}{\Delta_{\min}^3}N + \frac{\mu_{\max}}{\mu_{\min}}\cdot\frac{1}{\Delta_{\min}}N^{3/2}\right)\log(NT/\mu_{\min})\right)$$

When $\mu_1, \ldots, \mu_N$ are constants,

$$\mathbb{E}\left[R(T)\right] = O\left(\left(\frac{1}{\Delta_{\min}^2}N + \frac{\Delta_{\max}}{\Delta_{\min}^3}N + \frac{1}{\Delta_{\min}}N^{3/2}\right)\log(NT)\right).$$

Recall that $\mu_i \geq 1$ for all $i \in [N]$ and $\Delta_{\max} \leq 1$. Notice that the upper bound has a logarithmic dependence on $T$ when the gaps between $c_1\mu_1, \ldots, c_N\mu_N$ are fixed. On the other hand, the largest factor in $N$ is still $N^{2/3}$ as in the instance-independent upper bound (1).

## 4 Stochastic service times

What is described in Algorithm 2 is an extension of Algorithm 1 to the case of stochastic service times. We assume that for $i \in [N]$, the mean of job $i$'s service time is given by $T/\mu_i$ and known to the decision-maker. This incorporates the setting of deterministic service times as a special case. Unlike the deterministic case, some jobs may be finished during the preemption phase in the stochastic case.

---

**Algorithm 2** Preemptive-then-nonpreemptive empirical $c\mu$ rule for stochastic service times

---

**Require:** $T_s$ is a predetermined length of the preemption period. $J \leftarrow [N]$.
   **(Preemption phase)** For each $t = 1, \ldots, T_s$, select and process a job $j$ in $\arg\max_{i \in J} \hat{c}_{i,t}\mu_i$.
   If $j$ is completed, update $J \leftarrow J \setminus \{j\}$.
   **(Nonpreemptive phase)** While $J$ is nonempty, select a job $j$ in $\arg\max_{i \in J} \hat{c}_{i,t}\mu_i$ where $t$ is the current time step index.
   Then keep processing job $j$ until $j$ is completed and update $J \leftarrow J \setminus \{j\}$.

---

We prove that when the service time of each job is geometrically distributed, the expected regret of Algorithm 2 can be still sublinear in $T$ and subquadractic in $N$. The probability that each job $i$ is completed when it is served in a time slot is $\mu_i/T$. For this setting, we set $T_s$ as

$$T_s = \kappa \cdot N^{2/3}(T/\mu_{\min})^{2/3}\left(\log(NT/\mu_{\min})\right)^{1/3}$$

for some constant $\kappa > 0$. Based on the memoryless property of the geometric distribution, we obtain the following regret upper bound.

**Theorem 4.1.** *When the service time of each job $i$ is geometrically distributed with mean $\mu_i/T$, the expected regret of Algorithm 2 is*

$$\mathbb{E}\left[R(T)\right] = O\left(N^{5/3}(T/\mu_{\min})^{2/3}\left(\log(NT/\mu_{\min})\right)^{1/3}\right). \tag{3}$$

Compared to (1) and the upper bound given by Theorem 3.3 for the deterministic case, (3) has the same scaling in $T$ while it has an additional $N^{2/3}$ factor. Although we think that there is still a room for improvement in the regret analysis of Algorithm 2, it is interesting that the bound (3) already achieves the $T^{2/3}$ scaling and a subquadratic dependence on $N$.

## 5 Experiments

We run experiments to assess the numerical performance of the preemptive-then-nonpreemptive empirical $c\mu$ rule. We test the efficiency of Algorithm 1 compared to the preemptive empirical $c\mu$ rule

and the nonpreemptive version. We also evaluate the tightness of the proposed upper and lower bounds on regret by measuring how the expected regret behaves as a function of parameters $N$ and $T$ for randomly generated instances. Our code for running experiments and obtained data are publicly available in `https://github.com/learning-to-schedule/learning-to-schedule`. All experiments are conducted on an Intel Core i5 3GHz processor with 6 cores and 32GB memory.

Figure 1 shows the first set of results for comparing Algorithm 1 against the preemptive and nonpreemptive versions. For Figure 1, we used instances with $N = 20$, $T = 2000$, $\mu_i = 1$ for $i \in [N]$, and $c_1, \ldots, c_N$ being sampled from the uniform distribution on $[0.5 - \varepsilon, 0.5 + \varepsilon)$, where $\varepsilon$ is a parameter that we vary. For each value of $\varepsilon$, we generated 100 instances, and for each of which, we recorded

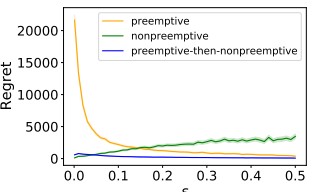 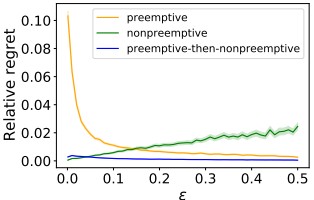

Figure 1: Comparing the three versions of the empirical $c\mu$ rule for the case of deterministic service times and equal service times: (left) regret and (right) relative regret.

the expected regret of each algorithm where the expectation is taken over the randomness in holding costs. The first plot in Figure 1 depicts how the (expected) regret changes by varying the value of $\varepsilon$, and the second plot shows the (expected) *relative* regret, defined as the regret divided by the minimum expected cumulative cost. As expected, the preemptive version suffers for instances of small $\varepsilon$ where the mean holding costs of jobs are close to each other, whereas the nonpreemptive $c\mu$ rule's regret does seem to increase for instances of large $\varepsilon$ where there may be large gaps between the jobs' mean holding costs. Compared to these two algorithms, our preemptive-then-nonpreemptive empirical $c\mu$ rule performs uniformly well over different values of $\varepsilon$. This trend continues even when jobs have

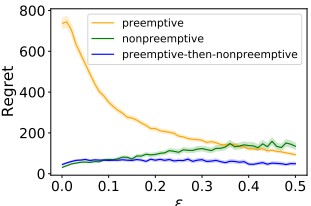 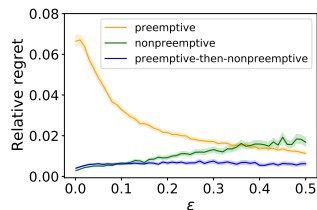

Figure 2: Comparing the three versions of the empirical $c\mu$ rule for the case of deterministic and heterogeneous service times: (left) regret and (right) relative regret.

heterogeneous service times. For the second set of results, we used the same setup as in the first experiment, but following [16], we sampled the mean service times $T/\mu_1, \ldots, T/\mu_N$ using a translated (heavy-tailed) Pareto distribution so that $T/\mu_i \geq 100$ for $i \in [N]$. More precisely, for each $T/\mu_i$, we sample a number $x_i$ from the distribution with the density function $f(x) = \frac{0.7}{x^{1.7}}$ for $x \in [1, \infty)$, and then, we set $T/\mu_i = 99 + \lfloor x_i \rfloor$. Here, the density function corresponds to the Pareto distribution with shape parameter $0.7$[1], which has infinite mean. As we assumed that each $T/\mu_i$ is an integer, we take $\lfloor x_i \rfloor$, to which we add 99 to ensure that $T/\mu_i$ is at least 100. Figure 2 shows that our algorithm achieves small regrets for all values of $\varepsilon$ even for the case of heterogeneous service times.

To examine how the expected regret of Algorithm 1 grows as a function of $T$, we test instances with $N = 20$ and different values of $T$ from 20 to 1,000,000. To understand how the expected regret depends on $N$, we also run experiment with instances with $T = 1000$ and different values of $N$ from 2 to 1000. For both experiments, we set $\mu_i = 1$ for $i \in [N]$ and $\varepsilon = 0.001$, and the reason for this choice is that the family of instances used for providing the regret lower bound (2) have jobs

---

[1]According to [16], Google's 2019 workload data shows that the resource-usage-hours, corresponding to the service times, of jobs follow the Pareto distribution with shape parameter 0.69 (see Figure 12 in [16]).

whose mean holding costs are concentrated around $1/2$ when $\mu_i = 1$ for $i \in [N]$. For each setup, we generated 100 random instances by sampling $c_1, \ldots, c_N$ from $[0.5 - \varepsilon, 0.5 + \varepsilon)$ uniformly at random. The left of Figure 3 is the log-log plot delineating the regret's dependence on $T$. The plot is

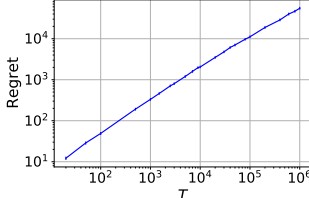 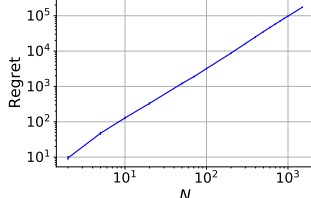

Figure 3: Examining how the regret grows as a function of $T$ (left) and $N$ (right).

almost linear, and its slope is roughly $3.4/4.9 \simeq 0.69$, which is close to the exponent $2/3$ for the $T$ factors in both the upper bound (1) and the lower bound (2). The right of Figure 3 is the log-log plot showing the regret's dependence on $N$. As the left one, the plot is also almost linear, and its slope is approximately $4.1/2.9 \simeq 1.41$. This result suggests that the upper bound (1) is close to being exact and that there may be a larger room for improving the lower bound (2).

## 6 Conclusion and future work

This paper proposes and studies the problem of finding a learning and scheduling algorithm to find a schedule of jobs minimizing the expected cumulative holding cost under the setting that statistical parameters defining individual jobs' cost functions are unknown a priori. We develop our algorithm as a learning-based variant of the well-known $c\mu$ rule for scheduling. The algorithm can be viewed as a combination of preemptive and nonpreemptive scheduling in the sense that it starts with a preemption phase that allows to learn enough about the jobs' cost functions, and then schedules jobs nonpreemptively. We provide a regret analysis of our algorithm where the benchmark is the minimum possible cumulative holding cost that is achievable only when the statistical parameters of jobs are fully known. We give bounds on the expected regret of our algorithm for both the case of deterministic service times and the setting of geometrically distributed stochastic service times. Lastly, we provide numerical results that support our theoretical findings.

Our theoretical analysis reveals that the algorithm achieves near-optimal performance guarantees, but the results rely on the assumption that the parameters $\mu_1, \ldots, \mu_N$ are constants. More precisely, when there are large gaps between the parameters, the upper and lower bounds provided in this paper have a large gap. Moreover, we also assumed that $\mu_i$'s are sufficiently small to ensure that the preemption period's length is smaller than the minimum service time of a job. Hence, we leave as an open question to provide matching upper and lower bounds on the expected regret of the preemptive-then-nonpreemptive empirical $c\mu$ rule for the case of large gaps in $\mu_1, \ldots, \mu_N$.

Another open question concerns the case of geometrically distributed stochastic service times. Although we have proved that the expected regret of our algorithm is sublinear in $T$ and subquadratic in $N$, we think that there exists a more refined regret analysis. Our argument is based on the observation that the jobs remaining after the preemption phase will have generated $T_s$ instantiated holding costs. However, as the service times of jobs are stochastic, the number of observations for a job is also a random variable, but we could not take this into account in our analysis.

One may also consider some variations of our problem by allowing for partial or delayed feedback. We can imagine a situation where the learner observes stochastic holding costs only for a subset of items in each time step, or another scenario is when there are some delays in receiving realized holding costs so that the learner observes only accumulated costs of jobs over multiple time steps. This may be of interest in real-world systems where only a limited information about stochastic holding costs is accessible by the learner due to computation / communication constraints in each time step.

Lastly, it is left for future work to study cases when *both* mean job holding costs and mean job service times are unknown parameters. [5] considers unknown mean service times, whereas our work studies the case of unknown mean job holding costs. Combining these two frameworks would be of interest.

## Acknowledgments and Disclosure of Funding

This research is supported, in part, by the Institute for Basic Science (IBS-R029-C1, Y2) and the Facebook Systems for ML Research Award.

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
