# OpenReview forum: "Scheduling jobs with stochastic holding costs"
_NeurIPS.cc/2021/Conference — NeurIPS 2021 Poster_

### Official Review · Reviewer_wNcX · 2021-07-12

**Rating:** 7
**Confidence:** 4

**Summary:**

The paper considers the problem where jobs of unknown weights and durations drawn from some distribution are available and need to be scheduled so as to minimize the sum of their weighted completion time (the paper calls this holding time). It is well known that when weights and durations are known then the so-called Smith's rule is optimal and the same is also the case when the durations used in Smiths rule are the "mean values" of the corresponding distribution(s).

Consider that the jobs can be preempted, i.e., the algorithm can choose at each time slot the job which Smiths rule would pick by using the aforementioned "mean" duration and the so-far learned information on the weight. Then in the worst case, all jobs would have weights that are close to each other, which would result in a -- more or less -- round robin schedule that completes all jobs close to the end of the schedule resulting in unnecessary delays, and in turn also regret.

On the other hand, an algorithm that non-preemptively picks a job and sticks to it, may select a job with an arbitrary weight first resulting in arbitrarily high regret.

The authors propose an algorithm that runs the preemptive schedule for some time-period before then switching to the non-preemptive one. The idea is that although, and as just described, at the extreme cases when the preemptive period has length either 0 or infinity the regret increases substantially, one can carefully choose a length so as to have learned enough information about the job weights by the end of that preemptive phase (intuitively, if the non-preemptive phase still processes two jobs in the wrong order then they have similar weights so it is not too bad), while at the same time not running the preemptive phase for too long. This results in an algorithm with sublinear regret.

**Ethical Concerns:**

I don't think there are any particular ethical concerns about this paper.

**Limitations And Societal Impact:**

Yes they have.

**Main Review:**

To the best of my knowledge the results presented in the paper are original, although they are in my opinion quite straightforward (which may actually be considered as an advantage since it results in a very simple algorithm).

Although the paper is general very well written, I feel that some parts can be substantially improved. In particular:

1) the model is not adequately described and self contained,
2) to my understanding the "learning" part is completely hidden in the supplementary material and by reading only the paper itself, it is not completely clear how and how effectively the weights learned, and
3) the sketches of the proofs although useful sometimes seem completely unrelated to the theorem statement. For example the sketch after Theorem 3.1 gives a two job example that essentially only implies that to keep regret as low as possible one needs to keep $T_s$ as low as possible (whereas the theorem statement naturally also considers the tradeoff between the opposing worst-case not mentioned in the "proof-sketch").

Most of these concerns are rectified though by looking into the supplementary material. Although I did not carefully go over every line in the proofs presented there, I skimmed over them and they seem to check out.

(Almost) matching lower bounds are also given, and experimental results are presented which are not too surprising (essentially the algorithm is compared to the only preemptive and only non-preemptive one which are the building blocks and the sublinear regret is depicted as well).

To summarize, on the positive side, the paper studies an interesting problem and gives a simple algorithm for it along with (almost) matching lower bounds), and the writing is in general very good. On the negative side, the paper is not completely self-contained without the supplementary material, and the algorithm is quite simple/straightforward.

**Time Spent Reviewing:**

4

---

> ### Author Response · Authors · 2021-08-09
> **Response to Reviewer wNcX**
>
> Thank you so much for your careful review and helpful comments. We have received constructive comments from 4 reviewers, and here are some common points raised:
>
> 1. Elaborate on applications and situations where the holding costs of jobs are unknown while job lengths are known.
> 2. Provide comparisons against prior works on the case where the job service time parameters are unknown while the job costs are deterministic.
> 3. Discuss the validity of assumptions imposed on our setting, e.g., static job list, sub-Guaissian holding costs, deterministic service times, and service times following a geometric distribution.
>
> We have treated these reviews with great care and given the corresponding responses.
>
> **Responses to comments**
>
> "To the best of my knowledge the results presented in the paper are original, although they are in my opinion quite straightforward (which may actually be considered as an advantage since it results in a very simple algorithm)."
> * Thanks for the positive feedback on the development of our algorithm!
>
> "Although the paper is general very well written, I feel that some parts can be substantially improved."
> * Thank you for the positive review and useful comments. We tried to provide proper responses and explanations to the comments and suggest ways to improve the parts mentioned.
>
> "The model is not adequately described and self contained."
> * We currently have a separate section in the introduction to define and describe our problem formulation. We have formally defined the notion of regret as a measure of analyzing the performance of a given algorithm. Perhaps, there is a need to add more details explaining the underlying scheduling problem, which is described in words. We will improve the part so that the reader can understand the model and problem formulation more clearly.
>
> "To my understanding the "learning" part is completely hidden in the supplementary material and by reading only the paper itself, it is not completely clear how and how effectively the weights learned."
> * The key learning part for our algorithm is to determine the switching point from preemptive to nonpreemptive scheduling and establishing regret upper bounds for this algorithm, and nearly matching regret lower bounds. This learning part is explained in our paper. Our primary goal is regret minimization -- "learning of weights" is only used for the purpose of regret minimization.
>
> "The sketches of the proofs although useful sometimes seem completely unrelated to the theorem statement. For example the sketch after Theorem 3.1 gives a two job example that essentially only implies that to keep regret as low as possible one needs to keep $T_s$ as low as possible (whereas the theorem statement naturally also considers the tradeoff between the opposing worst-case not mentioned in the "proof-sketch")."
> * The missing part in the proof sketch of Theorem 3.1 is that by Hoeffding's bound, $c_1-c_2$ can be bounded by $O(\sqrt{\log T/ T_s})$, which essentially gives rise to the result in the theorem statement. We can add this detail to improve the proof sketch of Theorem 3.1. As far as we are concerned, we only provide some high-level ideas or some key tools used due to the space limit for the other theorems. If required, we can include more details for proving the results.
>
> "To summarize, on the positive side, the paper studies an interesting problem and gives a simple algorithm for it along with (almost) matching lower bounds), and the writing is in general very good. On the negative side, the paper is not completely self-contained without the supplementary material, and the algorithm is quite simple/straightforward."
> * Thank you for the positive feedback on the results and writing of this paper. While revising our paper, we will improve the organization to make the paper more self-contained and better readable without referring much to the supplementary material.

---

### Official Review · Reviewer_jqaB · 2021-07-19

**Rating:** 5
**Confidence:** 4

**Summary:**

The paper presents algorithms for the problem of ordering jobs on a machine so as to minimize the cumulative holding cost. The assumption is that the holding cost in each timestep follows a sub-Gaussian distribution (i.i.d. across timesteps), but the the mean values of these distributions for individuals jobs are unknown to the algorithm. The lengths of the jobs are known upfront, or in an extension considered later in the paper, are generated from geometric distributions whose mean values are known upfront to the algorithm. If holding costs and job lengths were both known, then the optimal strategy is to process jobs in decreasing order of the ratio of the holding cost and job length. One natural strategy would be to implement this same rule, but with estimated mean holding costs based on previous iterations. This has the advantage of not committing to any job upfront, and therefore, being able to adjust to estimation errors of the holding costs. But, this strategy has the downside that if the holding costs are similar, then it can process the jobs round robin thereby keeping all jobs in the queue longer and incurring a large cost. The alternative is to commit to a job that has the largest value of this (estimated) ratio, and this strategy works well when the ratios are close to each other but not when they are well-separated, since estimation errors can be costly in the latter case. The algorithm proposed in the paper is natural combination of these ideas: start with a learning/exploration phase where you use the former strategy thereby not committing to any job but learning their holding costs, and then in the exploitation phase, commit to the job with the largest (estimated) ratio and run it to completion. In addition to obtaining regret bounds for this algorithm, the paper also gives lower bounds on regret that can obtained by any algorithm, and does numerical experiments to evaluate the performance in practice.

**Ethical Concerns:**

No ethical concerns as far as I can tell.

**Limitations And Societal Impact:**

The authors do a good job of spelling out their technical limitations, namely that the bounds are not tight unless one assumes the job lengths to be (scaled) constants. But, as I described above, a lot more discussion is required about the modeling limitations, and why the problem makes sense in some scenarios even with these limitations.

No negative societal impact as far as I can tell.

**Main Review:**

The technical work in this paper is quite good, but I am not convinced about the relevance of the specific problem model being studied here. What are situations where the holding costs are unknown but job lengths are known (at least the mean lengths are known)? The examples given at the beginning of the paper are about the relative importance (or urgency) of different jobs being unknown (and being fluid) in some situations, but they do not directly relate to unknown holding costs and known job lengths. For example, in a triage scenario, one should expect significant uncertainty in job lengths as well. Also, how general is the sub-Gaussian assumption on the holding costs? How about the assumption that if the job lengths are not known deterministically, then they should follow a geometric distribution? The paper does not make a convincing case that these are important scenarios to consider. Bereft of this explanation, the results obtained, while technically reasonable, have only limited significance in the research area. I also find the assumptions that all jobs arrive at the outset somewhat restrictive for the kind of applications being considered here. For instance, in data center scheduling or triage management, which are mentioned as motivating examples, the environment is clearly more dynamic and jobs actually arrive over time. The consideration of a static set of jobs is therefore a significant limitation of the paper. Finally, the connection with machine learning is also somewhat tenuous. One might argue that the algorithms are actually learning characteristics of the jobs over time and using them in the scheduling, but learning here basically means just observing samples and making simple estimates from them. This is a rather rudimentary form of learning. The relevance of the paper to machine learning, even in a broad sense, is limited.

I have read the authors' response (and appreciate the time and effort the authors spent in addressing all the points). The motivation of the paper is still the major point of concern. The applications described in the response appear retrofitted to the problem, rather than the other way around. The limitations, particularly the offline knowledge about jobs, also appear rather serious for the applications that this paper is targeted toward. For these reasons, my overall opinion about the paper remains unchanged.

**Time Spent Reviewing:**

1

---

> ### Author Response · Authors · 2021-08-10
> **Response to Reviewer jqaB**
>
> Thank you so much for your careful review and helpful comments. We have received constructive comments from 4 reviewers, and here are some common points raised:
>
> 1. Elaborate on applications and situations where the holding costs of jobs are unknown while job lengths are known.
> 2. Provide comparisons against prior works on the case where the job service time parameters are unknown while the job costs are deterministic.
> 3. Discuss the validity of assumptions imposed on our setting, e.g., static job list, sub-Guaissian holding costs, deterministic service times, and service times following a geometric distribution.
>
> We have treated these reviews with great care and given the corresponding responses.
>
> **Responses to comments**
>
> "The technical work in this paper is quite good"
> * Thank you for your positive feedback on the technical contributions of our paper.
>
> "What are situations where the holding costs are unknown but job lengths are known (at least the mean lengths are known)? The examples given at the beginning of the paper are about the relative importance (or urgency) of different jobs being unknown (and being fluid) in some situations, but they do not directly relate to unknown holding costs and known job lengths. For example, in a triage scenario, one should expect significant uncertainty in job lengths as well."
> * Traditional queueing system control literature assumes knowledge of various system parameters such as job holding cost parameters and job mean service times. In general, even with this knowledge, finding optimal or nearly optimal control policies is typically a non-trivial problem. Recent work started investigating queueing system control policies under uncertainty about some system parameters---for example, uncertainty about job mean service time parameters in the context of $c\mu$ rule scheduling. Our work considers a different problem where job holding costs are *stochastic* in a different way than what is typically considered in scheduling or queueing systems work, in which job holding costs are deterministic functions of stochastic job waiting times. In particular, our problem setting is more general than assuming job holding costs to be (deterministic) linear functions of stochastic job waiting times.
> * For example, consider an application scenario that arises in the context of content moderation in online social media platforms. In this application scenario, a job corresponds to reviewing a content item by a machine learning algorithm or a human reviewer. The holding cost of a content review job is a function of the number of accumulated views of the content item in the online social media platform, if the content item contains harmful content violating community integrity standards. In this application scenario, it is natural to model job holding costs by a stochastic process.
> * In our work we consider scheduling under stochastic job holding costs according to stochastic processes with independent and identically distributed increments with unknown mean values, under the assumption that mean job service times are known. This is similar to related work where mean job service times are assumed to be unknown parameters, and job holding costs are assumed to be known (in previous work assumed to be deterministic). It is left for future work to study cases when *both* mean job holding costs and mean job service times are unknown parameters. We will add a discussion of this and other interesting directions for future work in the paper.
>
> "Also, how general is the sub-Gaussian assumption on the holding costs? How about the assumption that if the job lengths are not known deterministically, then they should follow a geometric distribution? The paper does not make a convincing case that these are important scenarios to consider. Bereft of this explanation, the results obtained, while technically reasonable, have only limited significance in the research area."
> * We consider scheduling jobs with stochastic job holding costs. Our focus is on job holding costs according to stochastic processes with independent and identically distributed increments. The distribution of job holding cost increments is assumed to be from the family of sub-Gaussian distributions which allows for different parametric distributions, e.g. Bernoulli, Poisson, and Gaussian, which is quite general and may fit various application scenarios. As a side remark, note that assuming sub-Gaussian distributions is a common assumption for modeling stochastic rewards in multi-armed bandits (MAB) literature. Future work may consider job holding costs according to different stochastic processes, allowing for dependent increments.
> * Assuming that job service times are geometrically distributed (or exponentially distributed for models in continuous time) is a standard assumption in the context of queueing systems literature. This assumption is made for tractability of analysis, exploiting the memory-less property of geometric (exponential) distribution. In certain heavy-traffic limit regimes, this assumption is nonrestrictive.
>
> "I also find the assumptions that all jobs arrive at the outset somewhat restrictive for the kind of applications being considered here. For instance, in data center scheduling or triage management, which are mentioned as motivating examples, the environment is clearly more dynamic and jobs actually arrive over time. The consideration of a static set of jobs is therefore a significant limitation of the paper."
> * We referred to our motivating application scenarios to point out situations in which it is natural to model job holding costs by a stochastic process. In scheduling literature, it is common to consider a setting where either the set of jobs to be scheduled is given (possibly with release times) or job arrivals are assumed to be according to a stationary stochastic process (e.g. i.i.d. Bernoulli arrivals). Both problem formulations are interesting from a theoretical and a practical viewpoint. Understanding scheduling problems for a given set of jobs may provide useful insights into how to design scheduling algorithms for systems with dynamic job arrivals.
> * The scheduling problem we consider does assume that the input is a static list of jobs. The purpose of this paper is to lay down the first step toward understanding scheduling systems where job holding costs are stochastic with unknown mean cost values. We start with the most basic yet fundamental setting with a static set of jobs, for which we show that the famous $c\mu$ rule can be adapted with provable guarantees to the case when job holding costs are stochastic with unknown mean cost values---this adaptation requires switching from preemptive to nonpreemptive scheduling at a carefully selected switching time. Similar attempts can be made for the dynamic job arrival setting, into which our work provides insights.
>
> "Finally, the connection with machine learning is also somewhat tenuous. One might argue that the algorithms are actually learning characteristics of the jobs over time and using them in the scheduling, but learning here basically means just observing samples and making simple estimates from them. This is a rather rudimentary form of learning. The relevance of the paper to machine learning, even in a broad sense, is limited."
> * A key fundamental challenge for the design of learning algorithms in our setting stems from the definition of the reward function, which depends on the *order* of selected items, and the number of experiments per item is constrained. This is different from standard MAB problems where the reward function is a modular function of selected items (sum of mean rewards). In our problem formulation, samples are collected for each item in each time step, hence, using only "simple" estimates of mean job holding costs suffices. The learning question is still interesting as one needs to come up with algorithms that make online decisions by using these estimates. In contrast, in standard MAB problems, one typically uses more "sophisticated" indices representing items, accounting for the uncertainty of estimates and a different number of samples acquired across items (e.g. UCB indices) but then, typically, selections are made by the simple greedy algorithm using these indices.
> * One may consider variations of our problem formulation by allowing only for partial feedback---the learner observes stochastic holding costs only for a subset of items in each time step. This may be of interest in real-world systems where only a limited information about stochastic holding costs is accessible by the learner due to computation / communication constraints in each time step. These are some open research questions that may be of interest to the machine learning community. We will add this in our discussion of interesting directions for future research.

---

### Official Review · Reviewer_ub4m · 2021-07-19

**Rating:** 7
**Confidence:** 3

**Summary:**

Proposed work is considering a stochastic scheduling problem where we are given a single server and a set of jobs with unknown cost profile. It is well known that when the cost profile is known Smith's rule is optimal. Using this as a baseline, the authors define a natural regret measure. The central result is algorithm with sublinear regret and a roughly matching upper bound.

**Limitations And Societal Impact:**

Not applicable.

**Main Review:**


Proposed work is considering a stochastic scheduling problem where we are given a single server and a set of jobs with unknown cost profile. It is well known that when the cost profile is known Smith's rule is optimal. Using this as a baseline, the authors define a natural regret measure. The central result is algorithm with sublinear regret and a roughly matching upper bound.

The key observation is that neither preemptive or non-preemptive approach to scheduling jobs in a given time instant works in general. Indeed, a completely preemptive approach would delay jobs and non-preemptive approach is susceptible to estimation errors made in the beginning. Combining a preemptive approach with non-preemptive approach, the authors propose a natural algorithm where jobs are scheduled preemptively in the beginning and later non-preemptively. In a sense, it seems like an interesting use of exploration vs exploitation in the context of scheduling jobs.

In conclusion, a clean, well written paper with a concrete set of theoretical results. I vote for acceptance.

**Time Spent Reviewing:**

3 hours

---

> ### Author Response · Authors · 2021-08-09
> **Response to Reviewer ub4m**
>
> Thank you so much for your careful review and positive feedback. We have received constructive comments from 4 reviewers, and here are some common points raised:
> 1. Elaborate on applications and situations where the holding costs of jobs are unknown while job lengths are known.
> 2. Provide comparisons against prior works on the case where the job service time parameters are unknown while the job costs are deterministic.
> 3. Discuss the validity of assumptions imposed on our setting, e.g., static job list, sub-Guaissian holding costs, deterministic service times, and service times following a geometric distribution.
>
> We have treated these reviews with great care and given the corresponding responses.
>
> **Responses to comments**
>
> "In a sense, it seems like an interesting use of exploration vs exploitation in the context of scheduling jobs."
> * Thanks for this positive remark on the notion of exploration vs exploitation considered in our paper.
>
> "In conclusion, a clean, well written paper with a concrete set of theoretical results. I vote for acceptance."
> * We thank you for your time and effort in carefully reviewing this paper! Thanks for your positive feedback on the theoretical contributions and writing of our paper.

---

### Official Review · Reviewer_njGK · 2021-07-21

**Rating:** 6
**Confidence:** 3

**Summary:**

This paper studies a scheduling problem in which there are multiple jobs available in the system each with a different holding cost and service time. This scheduling problem is studied previously for the case that holding cost is deterministic and known and the service time is stochastic but unknown. This paper tackles the case in which holding cost is unknown and service time deterministic/stochastic and known. Then based on some interesting motivating examples, an algorithm is proposed that starts with preemptive scheduling and after a while, it switches to a non-preemptive mode. The regret of the proposed algorithm for both deterministic and stochastic service time is obtained and by characterizing the lower bound, the authors show their policy is near-optimal. Last, there is one additional instance-dependent regret analysis that is useful when some additional information is known about jobs. Numerical experiments clearly demonstrate the improved performance of the proposed algorithms is compared to full preemptive of full non-preemptive alternatives.

**Limitations And Societal Impact:**

yes

**Main Review:**

-- Overall, this paper tackles an interesting and novel scheduling problem. The new setting of unknown cost known service time is interesting and technically challenging as compared to the prior setting. The paper is well-written and clearly states its contribution and the technical part is polished and easy to follow. The only concern is about the significance of the results since it seems that the problem is very narrow and of less practical importance since typically cost is coming from the customer and is typically known while the service time highly depends on several unknown factors. Some comments are given below.

-- This paper assumes that the service times of jobs are known but their cost is unknown. However, this difference from the prior work is not motivated properly. The other case, i.e., unknown $\mu$, known $c$ makes sense in practice since jobs typically belong to a few categories with similar properties, however, their service time is random. The case of interest in this paper, however, is not clear how it makes sense. Also, there are some additional assumptions in the model that makes the problem less interesting from a practical perspective, e.g., availability of all jobs at the beginning, known $T$, etc. In short, the authors should justify their model both from a practical perspective and also from the added technical challenges as compared to the prior work and in particular [4,5].

Other comments: The title of the paper is misleading, scheduling is an extremely broad topic with a huge number of variants, and using learning for scheduling is also an active topic of research both in systems and theory communities. The scheduling problem in this paper is also very narrow and not so practical, so, the authors could change the title to better reflect the concrete topic of this paper.

-- It is not clear why the regret analysis for 2 jobs is given in the main body of the paper since the general case is also intuitive.

**Time Spent Reviewing:**

5

---

> ### Author Response · Authors · 2021-08-10
> **Response to Reviewer njGK**
>
> Thank you so much for your careful review and helpful comments. We have received constructive comments from 4 reviewers, and here are some common points raised:
>
> 1. Elaborate on applications and situations where the holding costs of jobs are unknown while job lengths are known.
> 2. Provide comparisons against prior works on the case where the job service time parameters are unknown while the job costs are deterministic.
> 3. Discuss the validity of assumptions imposed on our setting, e.g., static job list, sub-Guaissian holding costs, deterministic service times, and service times following a geometric distribution.
>
> We have treated these reviews with great care and given the corresponding responses.
>
> **Responses to comments**
>
> "Overall, this paper tackles an interesting and novel scheduling problem. The new setting of unknown cost known service time is interesting and technically challenging as compared to the prior setting. The paper is well-written and clearly states its contribution and the technical part is polished and easy to follow."
> * Thank you for the positive review on the novelty of our problem formulation, technical contributions, and the writing of the paper.
>
> "The only concern is about the significance of the results since it seems that the problem is very narrow and of less practical importance since typically cost is coming from the customer and is typically known while the service time highly depends on several unknown factors.
>
> This paper assumes that the service times of jobs are known but their cost is unknown. However, this difference from the prior work is not motivated properly. The other case, i.e., unknown $\mu$, known $c$ makes sense in practice since jobs typically belong to a few categories with similar properties, however, their service time is random. The case of interest in this paper, however, is not clear how it makes sense."
> * In scheduling and queueing systems literature, it is a common assumption that job service times are stochastic, e.g. following a geometric distribution, with mean job service times allowed to be job-class specific. In these works, it is commonly assumed that mean job service times are known. Recent work considered scheduling problems when mean job service time parameters are unknown. In all these works, it is assumed that job holding costs are according to deterministic job holding cost functions, e.g. linear (fixed marginal job holding costs) or non-linear, which are either known or unknown. Our problem formulation is fundamentally different in that we consider stochastic job holding costs where randomness is not (only) because of stochastic job waiting times (due to stochastic job service times) but because of other factors. For example, in content review platforms of online social media platforms, content item review jobs have costs driven by the number of accumulated views for a content item while the content item review job awaits to be served, if the content item contains harmful content. Note that we do allow for job service times to be stochastic, according to geometric distributions, which is a standard assumption in queueing systems control literature.
> * Some recent works from the operations research and management science community consider scheduling systems with unknown cost functions. For example, Alizamir et al. (2013) studied the optimal stopping problem for identifying the classes of jobs based on the realized waiting costs. Sun et al. (2018) and Levi et al. (2019) considered the problem of scheduling jobs with unknown costs and unknown service times, while their development is based on the assumption that testing a job discloses the exact values of its unit holding cost and service time. The main motivation to study their problem comes from medical triage scenarios where the "weights" of patients need to be learned. There are certainly lots of applications in practice where the costs, weights, and priorities of tasks are stochastic but unknown in advance.
>
> "Also, there are some additional assumptions in the model that makes the problem less interesting from a practical perspective, e.g., availability of all jobs at the beginning, known $T$, etc."
> * In our work we consider scheduling under stochastic job holding costs according to stochastic processes with independent and identically distributed increments with unknown mean values, under the assumption that mean job service times are known. This is similar to related work where mean job service times are assumed to be unknown parameters, and job holding costs are assumed to be known (in previous work assumed to be deterministic). It is left for future work to study cases when *both* mean job holding costs and mean job service times are unknown parameters. We will add a discussion of this and other interesting directions for future work in the paper.
> * In scheduling literature, it is common to consider a setting where either the set of jobs to be scheduled is given (possibly with release times) or job arrivals are assumed to be according to a stationary stochastic process (e.g. i.i.d. Bernoulli arrivals). Both problem formulations are interesting from a theoretical and a practical viewpoint. Understanding scheduling problems for a given set of jobs may provide useful insights into how to design scheduling algorithms for systems with dynamic job arrivals.
>
> "In short, the authors should justify their model both from a practical perspective and also from the added technical challenges as compared to the prior work and in particular [4,5]."
> * Our model is different from the prior work by Krishnasamy et al. (2018) in that our work focuses on stochastic job holding costs while the latter work considers the case of unknown service times only. We briefly mentioned in the paper that their setting requires the assumption that the number of classes is small, so as serving more jobs in a particular class, the characteristics of the class get revealed. In contrast, we do not make such an assumption, which means that each job can serve as a single class. In our case, we use the duration where a job stays in the system to learn the job's cost value.
> * Our problem formulation is also fundamentally different from the problem studied by Hsu et al. (2021). They consider a multi-class, multi-server queueing system setting where job-server assignments yield stochastic rewards -- one-off rewards independent of job waiting times. We instead consider the case where job *holding costs* are stochastic. These two problem formulations are rather different.
>
> "The title of the paper is misleading, scheduling is an extremely broad topic with a huge number of variants, and using learning for scheduling is also an active topic of research both in systems and theory communities. The scheduling problem in this paper is also very narrow and not so practical, so, the authors could change the title to better reflect the concrete topic of this paper."
> * If our paper gets accepted, we will change the paper's title so that it better reflects the problem we study - our candidate new title is "Scheduling jobs with stochastic holding costs".
>
> "It is not clear why the regret analysis for 2 jobs is given in the main body of the paper since the general case is also intuitive."
> * We use the special case of 2 jobs to explain the basic intuition of why a preemption phase is necessary and how to choose the length $T_s$ of the preemption phase. The core idea of having a preemption phase and switching to a nonpreemptive scheduling mode can be delivered with the case of 2 jobs, and at the same time, proving the corresponding regret upper bound result is quite straightforward. However, while we use the same underlying idea for the general case, it is more involved and technical to establish the regret upper bound result when there are more than 2 jobs -- this is because the order of jobs processed after the preemption phase needs to be considered.

---

### Decision · Program_Chairs · 2021-09-27

**Decision:**

Accept (Poster)

**Comment:**

The paper concerns online scheduling algorithm which aims at minimizing the expected cumulative holding costs incurred by jobs with a priori unknown parameters.

The reviewers found the problem tackled by the paper interesting and novel, and the framework of unknown costs challenging as compared to the prior settings. They liked the fact that the paper gives a simple scheduling algorithm whose performance (almost) matches lower bounds for the problem. The paper was also generally assessed as clear and well-written.

There were also some reservations raised, namely: the algorithm is rather straightforward, limited relevance of the specific problem considered (assumptions which might be too strong in practice).

However, the overall evaluation of the paper was mainly positive and thus I recommend the acceptance. I suggest that the authors incorporate the reviewers' remarks in the final version of the manuscript.